

**A study on some basic features of seiches, inertial**
**oscillations and near-inertial internal waves**
Shengli Chen, Daoyi Chen, Jiuxing Xing
Shenzhen Key Laboratory for Coastal Ocean Dynamic and Environment, Graduate School at
Shenzhen, Tsinghua University, Shenzhen 518055, China.
*Correspondence to*: Jiuxing Xing (jxx2012@sz.tsinghua.edu.cn)

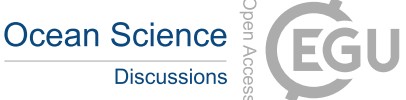

**Abstract**
Some basic features of seiches, inertial oscillations and near-inertial internal waves are investigated by
simulating a two-dimensional (x-z) shallow basin initialized by a wind pulse. Two cases with and without
the vertical stratification are conducted. For the homogeneous case, seiches and inertial oscillations
dominate. We find even modes of seiches disappear, which is attributed to a superposition of two seiches
generated at east and west coastal boundaries. They have anti-symmetric elevations and a phase lag of
nπ, thus their even modes cancel each other. The inertial oscillation shows typical opposite currents
between surface and lower layers, which is formed by the feedback between barotropic waves and inertial
currents. For the stratified case, near-inertial internal waves are generated are generated at land
boundaries and propagate offshore with increasing frequencies, which induce tilting of velocity contours
in the thermocline.
The inertial oscillation is uniform across the whole basin, except near the coastal boundaries (~20 km)
where it quickly declines to zero. This boundary effect is related to great enhancement of nonlinear terms,
especially the vertical nonlinear term ( $w\partial\mathbf{u}/\partial z$ ). With inclusion of near-inertial internal waves, the
total near-inertial energy has a slight change, with occurrence of a small peak at ~50 km, which is similar
to previous researches. We conclude that, for this distribution of near-inertial energy, the boundary effect
for inertial oscillations is primary, and the near-inertial internal wave plays a secondary role.











## 1. Introduction

If a water particle is subject to no force except the Coriolis force, it will move at the local inertial frequency (namely inertial motions). In reality, its frequency is usually slightly biased by other processes (Kunze, 1985). Near-inertial motion has been observed and reported in many seas (e.g. Alford et al., 2016;Webster, 1968). It is mainly generated by changing winds at the sea surface (Pollard and Millard, 1970;Chen et al., 2015b). The passage of a cyclone or a front can induce very strong near-inertial motions(D'Asaro, 1985), which can last for 1-2 weeks and reach a maximum velocity magnitude of 0.5-1.0 m/s (Chen et al., 2015a;Zheng et al., 2006;Sun et al., 2011). In deep seas, the near-inertial internal wave propagates downwards to transfer energy to depth (Leaman and Sanford, 1975;Fu, 1981;Gill, 1984;Alford et al., 2012). The strong vertical shear of near-inertial currents may play an important role in inducing mixing across the thermocline (Price, 1981;Burchard and Rippeth, 2009).

In shelf seas, near-inertial motions exhibit a two-layer structure, with an opposite phase between currents in the surface and lower layers (Malone, 1968;Millot and Crepon, 1981;MacKinnon and Gregg, 2005). By solving a two-layer analytic model using the Laplace transform, Pettigrew (1981) found this 'baroclinic' structure can be formed by inertial oscillations without inclusion of near-inertial internal waves. Therefore, due to similar vertical structure and frequencies, inertial oscillations and near-inertial internal waves are hardly separable, and could easily be mistakenly recognized as each other.

In shelf seas, the near-inertial energy increases gradually offshore, and reaches a maximum near the shelf break, found both in observations (Chen et al., 1996) and model simulations (Xing et al., 2004;Nicholls et al., 2012). Chen and Xie (1997) reproduced this cross-shelf variation both in linear and nonlinear simulations, and attribute it to large values of the cross-shelf gradient of surface elevation and the vertical gradient of Reynolds stress near the shelf break. By using the analytic model of Pettigrew (1981), Shearman (2005) argued that the cross-shelf variation is controlled by baroclinic waves which emanate from the coast to introduce nullifying effects on the near-inertial energy near shore. Kundu et al. (1983) found a coastal inhibition of near-inertial energy within the Rossby radius from the coast, which is attributed to the downward and offshore leakage of near-inertial energy near the coast. As many factors seems to have effects, the mechanism controlling the cross-shelf variation of near-inertial energy seems complicated.



During the occurrence of near-inertial motions, the wind force can also easily generate a seiche in close
or semi-close basins (de Jong, 2003). A seiche is a standing wave formed in an enclosed or partially
enclosed body of water, which has been widely observed in lakes, harbors, bays and seas (Miles,
1974;Metzner et al., 2000;Drago, 2009;Breaker et al., 2010). It has a period ranging from several minutes
to several hours. In some regions its amplitude can reach several meters (e.g.Wang et al., 1987), which
can induce flooding and cause damage to fishery and coastal facilities. In a closed rectangular basin of
length L and depth H, the seiche period is given by the Merian's formula:
$$T = \frac{2L}{n\sqrt{gH}} \tag{1}$$

where n=1, 2, 3, … is the mode number. Csanady (1973, using the Laplace transform) found that even
modes of seiches were absent. However, the absence of even mode seiches has not been reported in
observation or model simulations, probably due to irregular topography in reality which makes it difficult
to compute the exact period of each mode. And it is not known why even modes disappear from the
perspective of physics which we want to explore.
In this paper, we try to use simple simulations to investigate some basic properties of the inertial
oscillation and the near-inertial internal wave and differences between them. Generation of two-layer
structure of inertial oscillations and horizontal distribution of near-inertial energy are investigated in
details. The model is simple two-dimensional (x=600 km, z=60 m) and forced by a wind pulse with land
boundaries at both two sides. The missing of even mode seiches is also found and interpreted. Two cases
with and without vertical stratification are explored. Model settings are introduced in Section 2. In the
homogeneous case (Section 3), properties of seiches and inertial oscillations are investigated. In the
stratified case (Section 4), we study the difference near-inertial internal waves introduce. Summary and
discussion are presented in the final section (Section 5).
**2. Model Settings**
Numerical simulations are done by the MIT general circulation model (MITgcm) (Marshall et al., 1997).
The model is two-dimensional (i.e., the gradient along y is zero), with 3000 grids in the horizontal (x;
$\Delta x$ =200 m) and 30 grids in the vertical (z; $\Delta z$ =2 m). The water depth is uniform (60 m), with east and
west boundaries being closed (land). The vertical and horizontal eddy viscosities are assumed constants





as $5 \times 10^{-4}$ m$^2$/s and 10 m$^2$/s, respectively. The Coriolis parameter is $5 \times 10^{-5}$ s$^{-1}$ (at a latitude of 20.11 °N).
The model is forced by a spatially uniform wind which is kept westward and increases from 0 to 0.73
N/m$^2$ (corresponding to a wind speed of 20 m/s) for the first three hours and then suddenly stops. The
first case has no vertical stratification, while the second one has a stratification of two-layer structure
initially. Except stratification all settings of these two cases are the same.
**3.  Results without vertical stratification**
For the first case without vertical stratification, seiches and inertial oscillations are two dominant
processes.
**3.1 Seiches**
Due to the westward flow driven by the wind, the water level goes up at the west coast and down at the
east coast initially (Fig 1). A wave front propagates from each end at the speed of the barotropic wave
($\sqrt{gH}$ = 24 m/s or 87 km/h). As the wind stress and the water depth are uniform across the basin, the
elevation at the west is antisymmetric to that at the east (i.e. with the same amplitude but opposite phase).
The spectra of elevations is shown in Fig 2. At the inertial frequency, the elevation energy is slightly
increased. The most energetic peak is at the first mode of seiches, which is slightly biased by the earth
rotation effect. With the rotation, the wave frequency for each mode of seiche is given by (see the
Appendix):

$$\omega_n^2 = f^2 + \frac{n^2 \pi^2 gH}{L^2} \qquad (2)$$


where $f$ is the inertial frequency, $n$ the mode number, $g$ the gravity acceleration, $H$ the water depth, and
$L$ the basin width. As in most cases the horizontal scale of a closed basin is relatively small (<200 km),
the second term on the right hand side of Eq. (2) is much greater than the inertial frequency term, thus
the rotation effect is usually negligible. Here due to a large basin width (600 km), the rotation effect is
obvious.
The energy of the first mode is minimal at the middle of the basin (i.e., x=300 km) and maximal at two
boundaries. The second mode energy is almost negligible. The third mode, which has three nodes, is
much more energetic than the second mode. The fourth mode vanishes, while the fifth mode can be seen



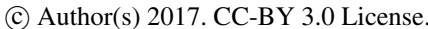


with five nodes although it has relative low energy. In a word, the even modes are absent. In the real
world, due to irregular topography, there is uncertainty in computing the exact period of each mode, and
the research on higher modes is limited. Csanady (1973, using the Laplace transform) found the even
modes of seiches absent. Here we propose an alternative way combining physics and mathematics to
interpret this phenomenon.
As derived from the appendix, the elevation of a seiche in a closed basin can be given by
$$\eta_1 = \cos\frac{n\pi x}{L}(A_1 \cos\omega_n t + A_2 \sin\omega_n t) \tag{3}$$

where A1 and A2 are arbitrary constants. As we see in Fig 1, a barotropic wave originates from the east
and west boundaries. Each barotropic wave can form a seiche. If the wind stress and the water depth are
spatially uniform, the elevation induced by the seiche at the west is antisymmetric to that at the east. The
wave generated at each end takes some time to reach the other end ($t_0 = L/\sqrt{gH}$), which causes a phase
difference of $n\pi$ between seiches driven at two ends (i.e., $\omega_n t_0 = 2\pi/T \times (L/\sqrt{gH}) = n\pi$). If the seiche
generated by the barotropic wave originating at the west boundary is denoted by (3), the seiche generated
by the wave from the east boundary is expressed as
$$\eta_2 = \cos\frac{n\pi x}{L}\left[-A_1 \cos(\omega_n t + n\pi) - A_2 \sin(\omega_n t + n\pi)\right] \tag{4}$$

The superposition of these two seiches is
$$\eta_1 + \eta_2 = \cos\frac{n\pi x}{L}\left[A_1 \cos\omega_n t\left(1 - \cos n\pi\right) + A_2 \sin\omega_n t\left(1 - \cos n\pi\right)\right] \tag{5}$$

The odd modes, i.e., n=1, 3, 5…, are amplified:
$$\eta_1 + \eta_2 = \cos\frac{n\pi x}{L}\left(2A_1 \cos\omega_n t + 2B_1 \sin\omega_n t\right) \quad (n=1,3,5...) \tag{6}$$

The even modes, i.e., n=2, 4, 6…, cancel each other:
$$\eta_1 + \eta_2 = 0 \quad (n=2,4,6...) \tag{7}$$

Therefore, even modes of seiches cancel each other.





### 3.2 Inertial oscillations

**153**

### 3.2.1 Vertical structures

**154**

**155** The model simulated velocities (Fig. 3) vary near the inertial period (34.9 hours). Since the vertical

**156** stratification is absent, this near-inertial motion is pure inertial oscillations. Spectra of velocities (not

**157** shown) indicate maximum peaks located exactly at the inertial period. The spectra of u also have a

**158** smaller peak at the frequency of the first mode seiche. As this simulation is two-dimensional, i.e., the

**159** gradient along $y$-axis is zero, the seiche has little energy in v which shows clearly regular variation at the

**160** inertial frequency.

**161** In the vertical direction, currents display a two-layer structure, with their phase being opposite between

**162** surface and lower layers. They are maximum at the surface, and have a weaker maximum in the lower

**163** layer (~40 m), with a minimum at the depth of ~20 m.   The velocity gradually diminishes to zero at the

**164** bottom due to the bottom friction. This is the typical vertical structure of shelf-sea inertial oscillations,

**165** which have been frequently observed (Shearman, 2005;e.g., MacKinnon and Gregg, 2005). In practice,

**166** this vertical distribution can be modified due to presence of other processes, such as the surface maximum

**167** being pushed down to the subsurface (e.g. Chen et al., 2013). Note that without stratification in this

**168** simulation the near-inertial internal wave is absent. However, this two-layer structure of inertial

**169** oscillations looks 'baroclinic', which makes it easy to be mistakenly attributed to the near-inertial internal

**170** wave (Pettigrew, 1981).

**171** It is interesting that currents of non-baroclinic inertial oscillations reverse between the surface and lower

**172** layers. This is usually attributed to the presence of the coast, which requires the normal-to-coast transport

**173** to be zero, thus currents in upper and lower layers compensate each other (Millot and Crepon, 1981;Chen

**174** et al., 1996). Here we try to give more detail on how this process works.

**175** As the westward wind blows for first three hours, the initial inertial current is also westward and only

**176** exists in the very surface layer (Fig. 4). In the lower layer there is no movement initially. Thus a westward

**177** transport is produced, which generates a rise (in the west) and fall (in the east) of elevation near coastal

**178** boundaries. The elevation slope behaves in a form of barotropic wave which propagates offshore at a

**179** great speed (87 km/h). The current driven by the barotropic wave is eastward, and uniform vertically.

**180** Therefore, with arrival of the barotropic wave the westward current in the surface is reduced, and the




movement in the lower layer commences (Fig. 4). After passage of the first two barotropic waves
(originated from both sides), currents in the lower layer have reached a relatively great value, while
currents in the surface layer have been largely decreased (Fig. 5a). Accordingly, the depth-integrated
transport diminishes a lot. This is like a feedback between inertial currents and barotropic waves. If only
the depth-integrated transport of currents exist, barotropic waves will be generated, which reduce the
surface currents but increase the lower layer currents, thus reduces the current transport. It will end up
with inertial currents in the surface and lower layers having opposite directions and comparable
amplitudes. As seen from Fig. 3b, the typical vertical structure of inertial currents is established within
the first inertial period. At a place further offshore, such as at x=200 km, the barotropic wave takes about
two more hours to arrive compared with x=70 km, thus the maximum value of inertial currents in lower
layer is lagged behind that in surface layer (Fig. 5b).
**3.2.2 Horizontal distributions of inertial energy**
The inertial velocities are almost entirely the same across the basin (Fig. 6), except near the boundary.
This indicates that inertial oscillations have a coherence scale of almost the basin width. This is because
in our simulation the wind force is spatially uniform, and the bottom is flat. The inertial velocities in the
lower layer have slightly more variation across the basin than those in the surface layer, because inertial
velocities in the lower layer is correlated to propagation of barotropic waves as discussed in 3.2.1, while
the surface inertial currents are driven by spatially uniform wind. In shelf sea regions, the wind forcing
is usually coherent as the synoptic scale is much larger, however, the topography that is mostly not flat
could generate barotropic waves at various places, and thus significantly decrease coherence of inertial
currents in the lower layer.
The spectra of velocities in the inertial band are almost uniform except near the boundaries (Fig. 7),
consistent with the velocities. Near the boundaries, the inertial energy declines gradually to zero from
x=~20 km to the coast wall. The east side has slightly greater inertial energy and a bit wider boundary
layer compared to the west side.
We calculate the nonlinear and inertial terms in the momentum equation and find that nonlinear terms
are of relatively high values initially within 2 km away from the land boundary (Fig 8bc), where the
inertial term is smaller (Fig 8a). For the time-averaged values (Fig 8d), the vertical nonlinear term is two



times more than the horizontal nonlinear term. The inertial term drops sharply near the boundary, and
rises gradually with the distance away from the boundary. At x> 15km, it keeps an almost constant value
which is much greater than nonlinear terms. Thus it is concluded that the significant decrease of inertial
oscillations near the boundary is due to influence of nonlinear terms, especially the vertical nonlinear
term.
**4.   Near-inertial internal waves**
In addition to inertial oscillations, near-inertial internal waves are usually generated along when the
vertical stratification is present. However, due to their close frequencies inertial oscillations and near-
inertial internal waves are difficult to be separated. Thus we run a second simulation with the presence
of stratification to investigate differences that near-inertial internal waves introduce. The temperature is
20 ℃ in the upper layer (-30 m<z<0), and 15 ℃ in the lower layer (-60 m<z<-30 m). The salinity is
constant, so the density is determined by the temperature.
**4.1 Temperature distributions**
Fig. 9 shows the evolution of temperature profiles with time. One can see an internal wave packet is
generated at the west coast, and then propagates offshore. The wave phase speed is around 1 km/h,
consistent with the theoretical value computed using the stratification. Before arrival of internal waves,
the temperature at mid-depth diffuses gradually due to vertical diffusion in the model. For a fixed position
at x=20 km (Fig. 10), the temperature varies with the inertial period (34.9 hours) and the amplitude of
fluctuation declines gradually with time. At x=60 km and x=100 km, the strength of internal waves is
much reduced. And wave periods are shorter initially, followed by a gradually increase to the inertial
period. At x=140 km, the internal wave becomes as weak as the background disturbance.
A spectral analysis of the temperature at mid-depth (z=-30 m) is shown in Fig. 11a. The strongest peak
is at near the inertial frequency (0.69 cpd), but only confined to the region close to the boundary (x<40
km). In the region 20km<x<70km, the energy is also large at higher frequencies of 0.8-1.7 cpd. This
generally agrees with properties of Poincaré waves. During a Rossby adjustment, the waves with higher
frequencies propagate offshore at greater group speeds, thus for places further offshore the waves have
higher frequencies (Millot and Crepon, 1981). While the wave with a frequency closest to the inertial
frequency moves at the slowest group velocity, and it takes a relatively long time to propagate far offshore,

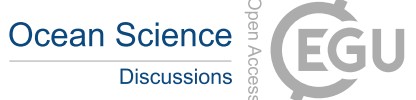

thus it is mostly confined to near the boundary. By solving an idealized two-layer model equation, the
response of Rossby adjustment can be expressed in form of Bessel functions (Millot and Crepon,
1981;Gill, 1982;Pettigrew, 1981), as in Fig. 11cd showing the spectra of mid-depth elevation. The
difference from our case is obvious. The frequency of theoretical near-inertial waves increase gradually
with the distance from the coast, while in our case this property is absent. And the theoretical inertial
energy has a e-folding scale of almost the Rossby radius (54 km), while in our case the e-folding scale is
much smaller (~15 km).
**4.2 Velocity distributions**
With presence of near-inertial internal waves, the contours of velocities near the thermocline tilt slightly
(Fig. 12d), and indicates an upward propagation of phase, thus a downward energy flux. This can also be
seen in vertical spirals of velocities (Figs. 12e and 12f). With only inertial oscillations, current vectors
mostly point toward two opposite directions. Once the near-inertial wave is included, the current vectors
gradually rotate clockwise with depth.
The spatial distribution of the near-inertial energy is also slightly changed compared to the case with only
inertial oscillations (Fig. 13 and Fig. 7). It is also greatly reduced to zero in the boundary layer (0-20 km)
like the case without stratification. But at ~50 km away from the boundary the inertial energy reaches a
peak. Further away (>100 km) it becomes a constant. This spatial distribution of inertial energy is similar
to that observed in shelf seas, with maximum near the shelf break (Chen et al., 1996;Shearman, 2005).
In our case, the boundary layer effect which induces a sharp decrease to zero makes a major contribution,
and near-inertial internal waves which bring a small peak further offshore make a secondary influence.
**5.   Summary and discussion**
Two sets of idealized simple two-dimensional (x-z) simulations are conducted to examine the response
of a shallow closed basin (600 km×60 m) to a wind pulse. The first case is homogeneous, in which
properties of seiches and inertial oscillations are investigated. Barotropic waves are generated at two
coastal boundaries which then propagate and reflect to form seiches. The seiche has the horizontal
structure and frequencies consistent with the theory. Seiches of even-number modes are absent, which
has been rarely reported. By using the Laplace transform to solve the equations, Csanady (1973) found
even mode seiches absent. We interpret it as superposition of two seiches, which are formed by barotropic



waves originating from east and west boundaries. They have anti-symmetric elevation and a phase lag
of nπ, thus their even mode cancels each other. The mechanism we propose is more physical, and thus a
good supplement to explain this phenomenon. Note that for the even modes to be absent the wind forcing
and the topography are required to be uniform spatially to keep these two seiches having anti-symmetric
elevation.
The inertial oscillation is energetic in the homogeneous case. It has a two-layer structure, with currents
in the surface and lower layers being opposite in phase, which has been reported frequently in shelf seas.
We find that the inertial current is confined in the surface layer initially. The induced depth-integrated
transport generates barotropic waves near boundaries which propagates quickly offshore. The flow
driven by the barotropic wave is independent of depth and opposite to the surface flow. Thus the surface
flow is reduced but the flow in the lower layer is increased, as a result the transport diminishes. This
feedback between barotropic waves and currents continues and ends up with the depth-integrated
transport vanishes, i.e., inertial currents in upper and low layers having opposite phases and comparable
amplitudes. In our simulation, within just one inertial period the typical structure of inertial currents has
been established. By solving a two-layer analytic model using the Laplace transform, Pettigrew (1981)
also found the vertical structure of opposite currents associated with inertial oscillations. He argued that
the arrival of a barotropic wave for a fixed location cancels half of the inertial oscillation in the surface
layer, and initiates an equal and opposite oscillation in the lower layer. Our simulations further
demonstrate the role of barotropic waves in forming this feature, and shows some more realistic details
during this process.
The second case is set up with idealized two-layer stratification, thus near-inertial internal waves are
generated. For a fixed position, velocity contours clear obvious tilting near the thermocline, and velocity
vectors display clearly anti-cyclonic spirals with depth. These could be useful clues to examine
occurrence of near-inertial internal waves. Near the land boundary the vertical elevation generates
fluctuations of thermocline that propagate offshore. The energy of near-inertial internal waves is confined
to near the land boundary (x< 40 km). At positions further offshore, the waves have higher frequencies.
This is generally consistent with properties of Rossby adjustment process. However, our simulated result
also shows evident discrepancies from theoretical values obtained in classic solutions of Rossby
adjustment problem.





The inertial oscillation has a very large coherent scale of almost the basin scale (600 km). It is uniform
in both amplitude and phase across the basin, except near the boundary (~20 km offshore). The energy
of inertial oscillations declines gradually to zero from x=20 km to the coast. This boundary effect is
attributed to influence of nonlinear terms, especially the vertical term ( $w\partial\mathbf{u}/\partial z$ ), which are greatly
enhanced near the boundary, and overweighs the inertial term ( $f\mathbf{u}$ ). When near-inertial internal waves
are produced in the stratified case, the distribution of total near-inertial energy is modified slightly near
the boundary. A small peak appears at ~ 50 km offshore. This is similar to the cross-shelf distribution of
near-inertial energy observed in shelf seas (Chen et al., 1996;Shearman, 2005). This energy distribution
has been attributed to different reasons, such as downward and offshore leakage of near-inertial energy
near the coast (Kundu et al., 1983), the variation of elevation and Reynolds stress terms associated with
the topography (Chen and Xie, 1997) and the influence of the baroclinic wave (Shearman, 2005;Nicholls
et al., 2012). In our simulations, this horizontal distribution of near-inertial energy is primarily controlled
by the boundary effect on inertial oscillations, and the near-inertial internal wave makes a secondary
effect.



**Acknowledgements**
We are very appreciated for comments from John Huthnance. This study is supported by the National
Basic Research Program of China (2014CB745002, 2015CB954004), the Shenzhen government
(201510150880, SZHY2014-B01-001), and the Natural Science Foundation of China (41576008,
41276006, U1405233, and 40976013). Shengli Chen is sponsored by the China Postdoctoral Science
Foundation (2016M591159).







320               **Appendix: the derivation for the general solution of seiches**

The governing equations for seiches can be simplified as:
$$\frac{\partial u}{\partial t} - fv = -g\frac{\partial \eta}{\partial x} \tag{A1a}$$

$$\frac{\partial v}{\partial t} + fu = 0 \tag{A1b}$$

$$\frac{\partial \eta}{\partial t} + H\frac{\partial u}{\partial x} = 0 \tag{A1c}$$

where u and v are eastward and northward velocities, $\eta$ the elevation, $f$ the inertial frequency, $g$ the gravity
acceleration, $H$ the water depth. Substitutions of $\eta$ and $v$ by $u$ give
$$\frac{\partial^2 u}{\partial t^2} + f^2 u - gH\frac{\partial^2 u}{\partial x^2} = 0 \tag{A2}$$

If we assume
$$u = X(x)T(t) \tag{A3}$$

and substitute (A3) in (A2), we get
$$\frac{T''}{T} + f^2 = \frac{gHX''}{X} \tag{A4}$$

If a function of $t$ equals a function of $x$, they have to both equal a constant,
$$\frac{T''}{T} + f^2 = \frac{gHX''}{X} = -C \tag{A5}$$

The equation of $x$ is then given by
$$gHX'' + CX = 0 \tag{A6}$$

The solution of (A6) can be readily obtained





$$X = C_1 \sin \sqrt{\frac{C}{gH}} x + C_2 \cos \sqrt{\frac{C}{gH}} x \quad (C > 0) \tag{A7}$$

where $C_1$ and $C_2$ are arbitrary constants. The across-coast velocity must vanish at boundaries, i.e., u=0 at
x=0, L, thus
$$X = C_1 \sin \sqrt{\frac{C}{gH}} x \tag{A8a}$$

$$C = \frac{n^2 \pi^2 gH}{L^2} \tag{A8b}$$

The solution for $T$ is then
$$T = C_3 \sin \omega_n t + C_4 \cos \omega_n t \tag{A9a}$$

$$\omega_n^2 = f^2 + \frac{n^2 \pi^2 gH}{L^2} \tag{A9b}$$

where $C_3$ and $C_4$ are arbitrary constants. Therefore the solution for $u$ is
$$u = C_1 \sin \frac{n\pi x}{L} \left( C_3 \sin \omega_n t + C_4 \cos \omega_n t \right) \tag{A10a}$$

and the solutions for of $\eta$ and $v$ are
$$\eta = C_1 \frac{n\pi H}{\omega_n L} \cos \frac{n\pi x}{L} \left( C_3 \cos \omega_n t - C_4 \sin \omega_n t \right) \tag{A10b}$$

$$v = C_1 \frac{f}{\omega_n} \sin \frac{n\pi x}{L} \left( C_3 \cos \omega_n t - C_4 \sin \omega_n t \right) \tag{A10c}$$







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





**Figures**


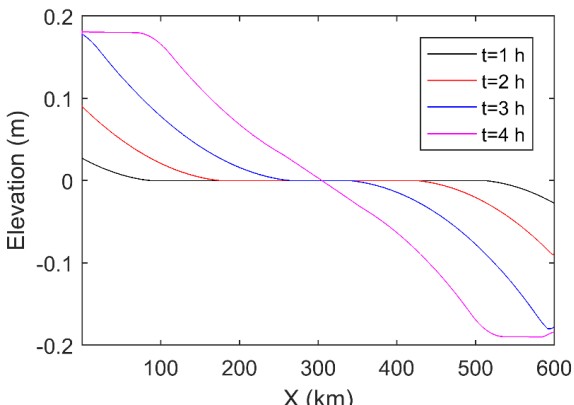


Fig 1. Elevation varying in the first 4 hours.


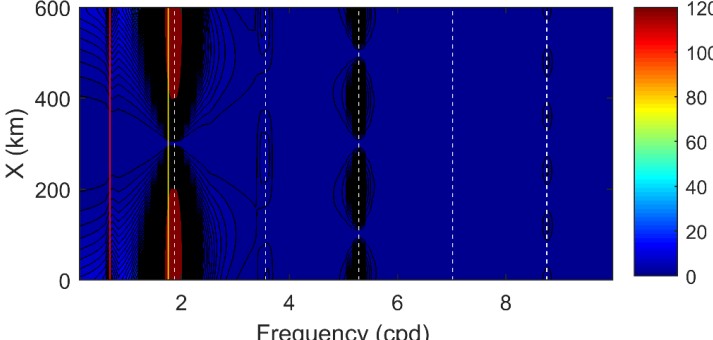


Fig 2. Elevation power spectral density (m$^2$s) dependence with x direction. The red line denotes the

inertial frequency, and the yellow line is the frequency of first mode seiche without the rotation effect.

The white dash lines are frequencies of first five modes of seiches computed by Eq. (2). The contour

interval is 1 m$^2$s.


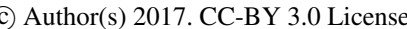



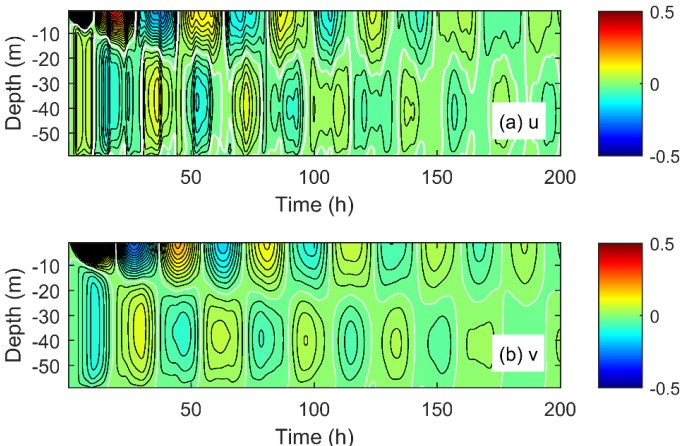


Fig. 3 Velocities (u and v, m/s) at x=70 km. The white lines denote the value of zero. The contour

interval is 0.02 m/s for both panels.


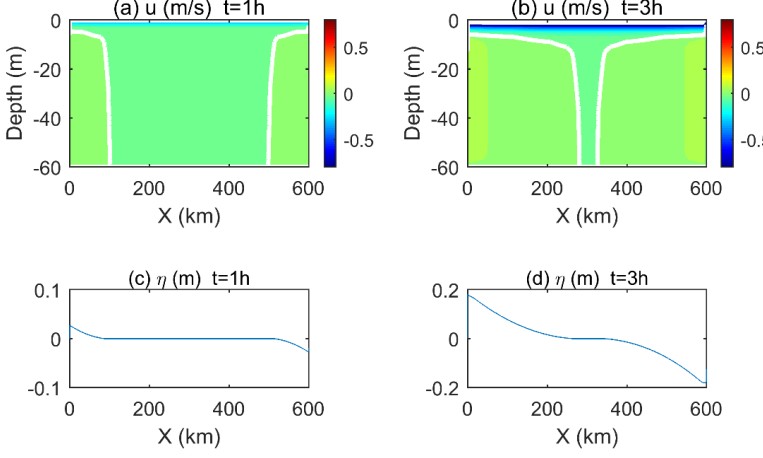


Fig. 4 Snapshots of eastward velocity and elevation ($\eta$) at t=1 and 3 hour. The white lines represent the

value of zero.




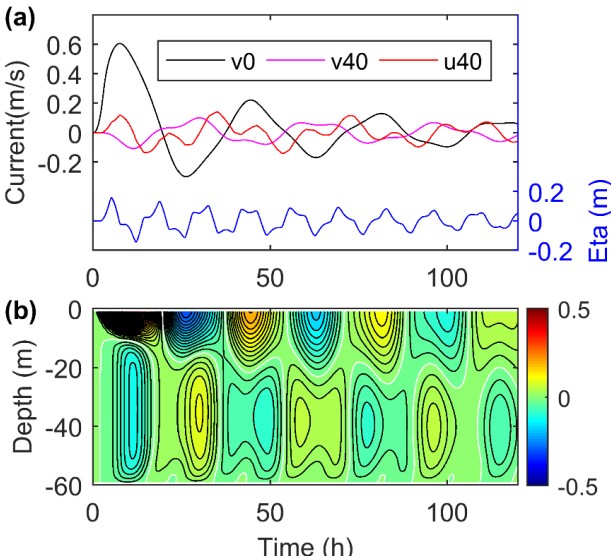

Fig. 5 (a) Time series of velocities and elevation at x=200 km. 'v0' and 'v40' mean the northward

velocity (v) at depths of 0 m and 40 m, and 'u40' is the eastward velocity (u) at 40 m. (b) Contours of v

at x=200 km. The white lines denote the value of zero, and the contour interval is 0.02 m/s.

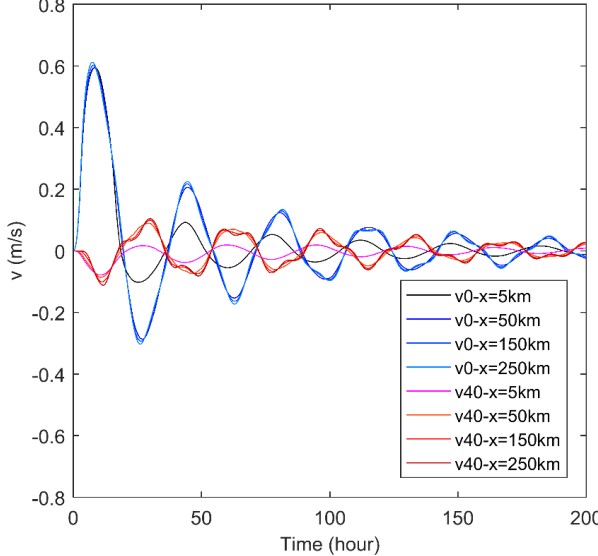









463   Fig. 6 Time series of the northward velocity (v) at different depths and positions. 'v0' and 'v40' mean v

464         at depths of 0 m and 40 m.



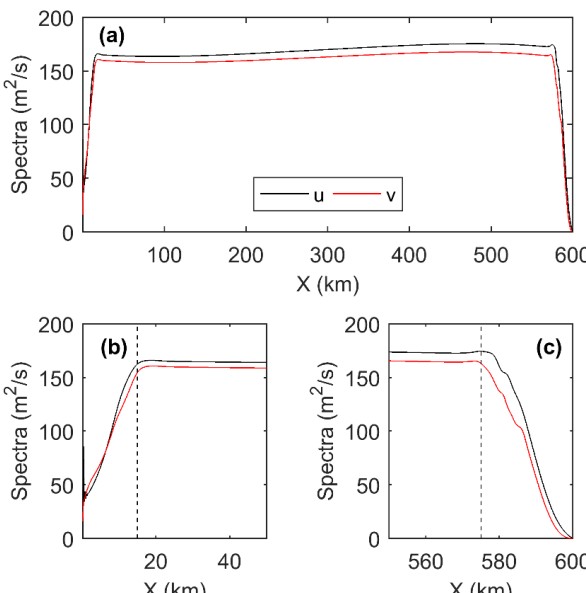


468   Fig. 7 Spatial variation of spectra of velocities in near-inertial band for the homogeneous case. (b) and

469        (c) display detailed values near boundaries.






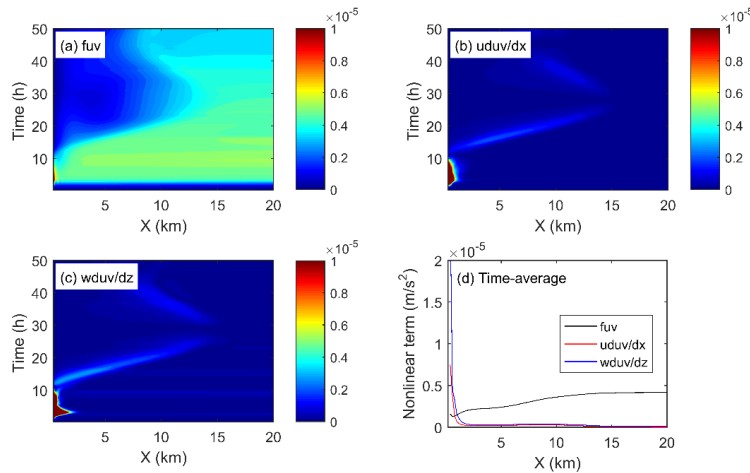


Fig 8. Variation of depth-mean inertial and nonlinear terms (m/s$^2$). The inertial term (a) is calculated as
$\left| f\left(u+iv\right) \right|$, the horizontal nonlinear term (b) is $\left| u\left(\partial u/\partial x + i\partial v/\partial x\right) \right|$, and the vertical nonlinear
term (c) is $\left| w\left(\partial u/\partial z + i\partial v/\partial z\right) \right|$. (d) Time averaged value for the first 50 hours.

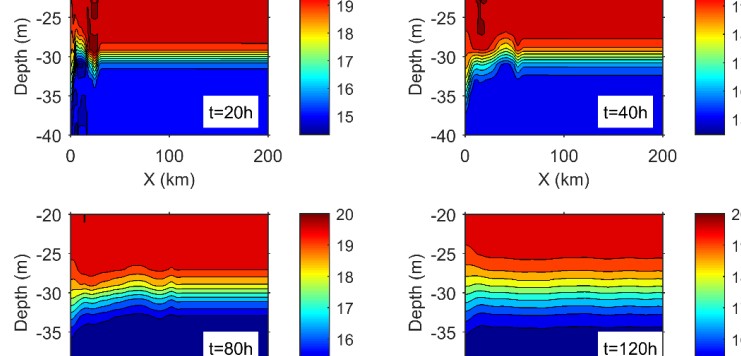


Fig. 9 Snapshots of temperature profiles at t=20h, 40h, 80h and 120h. The contour interval is 0.5 ℃.



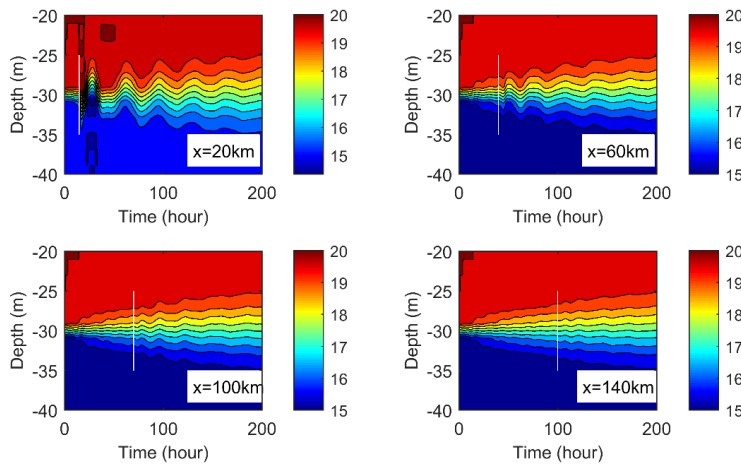


Fig. 10 Time series of temperature at x=20, 60, 100 and 140 km. White lines denote arrival of internal

481                    waves. The contour interval is 0.5 ℃.


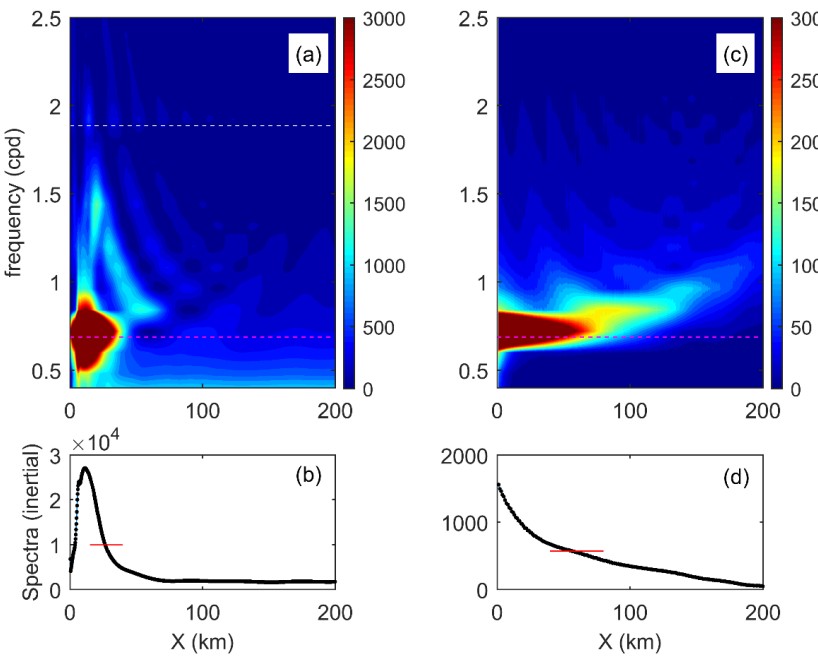


Fig. 11 (a) Spectra of the temperature at the mid-depth (z=-30 m). The pink dash line represents the



inertial frequency, and the white line is the first mode seiche frequency. (b) Sum of spectra in inertial
band with a red line denoting the e-folding value of the peak. (c) Theoretical spectra of mid-depth
elevation calculated from the solution in the form of a Bessel function as in Eq. 3.16 of Pettigrew
(1981). (d) Same as (b) but for theoretical spectra.

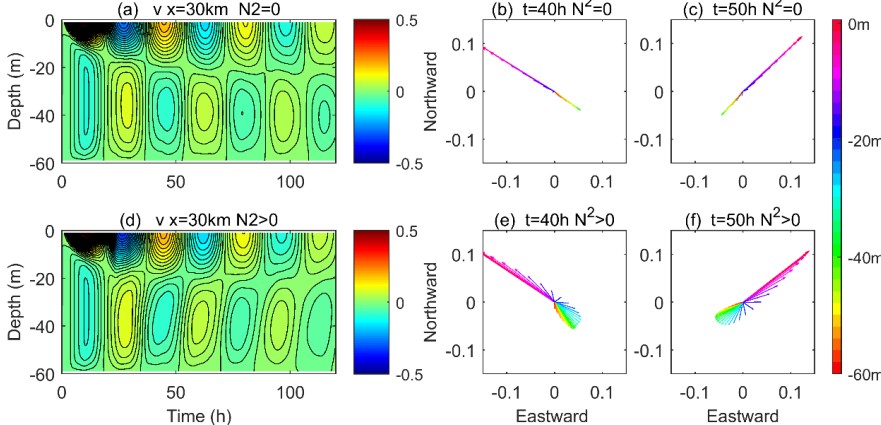


Fig. 12 Distribution of velocity v (m/s) and current spirals for the cases without (a, b, c) and with (d, e,
f) stratification at x=30 km. The contour interval is 0.02 m/s.





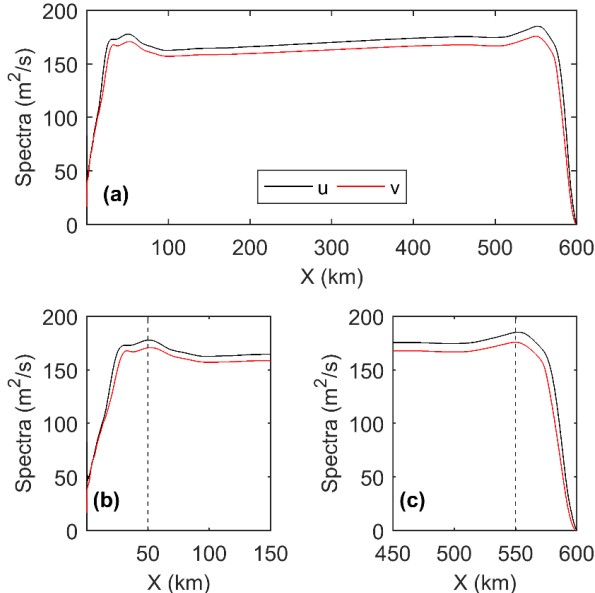


Fig. 13 Spatial variation of spectra of velocities in near-inertial band for the stratified case. (b) and (c)

496                      display detailed values near boundaries.