# Peer review of "A study on some basic features of inertial oscillations"

_Ocean Science, 2017_

## Referee Comment (RC1) · Anonymous Referee #1 · 21 Jun 2017

Aim of study is to investigate the absence of hypothesised even numbered modes of osculation within standing wave systems - called Seiches. Using MITGCM for an idealised basin 600km long (60m water depth) this is studied. The motivation, hypothesis and conclusion appears to be unclear, and needs to be stated; Further, the writing style also needs to be improved and much more detail and confidence in reported results are needed in my opinion. For example, a conclusion is needed in which the authors should state why study this phenomena (Why is this important?)? What was found? What this means for the scientific community (and stake holders)? Below are some further suggested changes to the text, but this is not a full list. Suggested improvements to text: Abstract - add more detail to explain why this study is about, why important, what was one and what this means. L25 - what kind of basin? an idealised rectangular? cer-

tainly appears large enough for Coriolis (inertial oscillations) How was it studied? (i.e. a model). L26 - what is "vertical stratification" I think you mean "stratified" L27 what are "even modes" explain further please to aid reader L53 (and throughout document) - please add space between references - for example here, "2016;Webster" should be "2016; Webster" L55 "motions(D'.." please add space and check throughout document L90 - 91. Nice clear aim, but why is this important? What is your hypothesis? Please be clear. L100 - So you use the MITGCM model. Please add much more detail about this model both here and in the introduction. Why use this model? How do you know the model is correct? What are the discretised equations? How is wind stress parameterised into the model? Are the boundary conditions non-slip? What density of water is assumed? Furthermore, how can you be confident of the model results? I am sure this is a classical problem and could be compared to other models and theory for example (i.e. I could be mean here and ask if you used a different model would you get the same result?). Lastly, what stratification is used in the second case? what temperature, at what depth is thermocline?

---

## Author Comment (AC1) · 27 Jun 2017

We are very appreciated for your comments, which will definitely help us to improve this manuscript much. Following your suggestions, we will add sentences to state the motivation, hypothesis and conclusions much more clearly, and show more detailed results. You also mentioned a list of some missing details in the abstract and the text, we will add them in the new manuscript. We will go throughout the whole manuscript to modify such deficiencies. All changes will be updated in a new version of the manuscript, along with a detailed response to your comments.
* * *

---

## Referee Comment (RC2) · Anonymous Referee #2 · 18 Jul 2017

The authors set out to describe inertial oscillations in a lake (or a cross section of an embayment – I'm not quite sure which). We are never told why they do this, or what the overall aim/hypothesis is, or what new knowledge we are going to get. The writing is sketchy and needs to be improved. However, at the end of the day, I don't think there is much new in this study, and I will outline why below.

More specific comments: L60: it is only 2-layer if the stratification is 2-layer. Please clarify/amend. Introduction: what is the purpose of this study? What research question/hypothesis will be answered/tested? Why? How does the set-up used relate to a real world scenario? Section 2: The model description is far too brief and sketchy. Why was the specific geometry chosen and how does that relate to the ocean? With closed ends it is more like a lake, unless we are looking at a cross section. Also, 2m vertical

[Figure]

resolution seems quite coarse for the study. We are never told what the 2-layer stratification look like (i.e., layer depths and densities). How long are the simulations? Why the chosen magnitude of the wind? It must also be noted that the barotropic Rossby radius is almost 500 km for the current set-up, so there will be interactions within the basin which must be quantified. Section 3.1: The discussion about seiches doesn't anything new to the field, and the analytical solution can be found in textbooks. That even mods cancel out is quite fundamental and not a novel result (and it doesn't require a numerical model simulation to be shown). L175 and onwards: Again, this is nothing new and the mechanism is already described by the authors: the flow must balance and cancel, hence the vertical structure. The description is fundamental and doesn't give any new insights – the same conclusions can be drawn from, e.g., the text by Csanady (1982). L223-224: show the computation and use SI units. L242-243: why is your case different from theory? Appendix: this is textbook stuff and can be omitted. Instead a reference can be added.

---

## Author Comment (AC2) · 19 Jul 2017

We are very appreciated for your detailed comments. We admit that the writing is too sketchy, so that many aspects have not been properly described, and many details are missing. We decide to clarify them in the response to your comments here at first, and will include them with more details in the new version of manuscript. The purpose of this study is:

1. Discuss the cross-shelf distribution of near-inertial energy. As mentioned in the Introduction (Lines 66-76), observations show the near-inertial energy is maximum near the shelf break, from which it declines onshore and offshore. There has been a couple of research on discussing it, but no common interpretation is reached. Nicholls

et al. (2012) simulated the near-inertial motion in the Caspian Sea, and found that the decrease of near-inertial energy depends on the distance from the coast, rather than correlated with the water depth. By solving the analytical model with a constant water depth, Pettigrew (1981) argued the near-inertial wave is responsible for the decline of near-inertial energy near the coast. Therefore, we set up a simulation with a constant water depth to explore their ideas. The water depth is shallow (60 m) as in the shelf seas. The basin is chosen to be very wide (600 km) to guarantee that the near-inertial waves generated at one end do not reach the other end during simulated duration. In our simulation, a gradual offshore increase of near-inertial energy is present in the case with only inertial oscillations. A small peak of near-inertial energy can be seen at ∼50 km offshore in the case including the near-inertial waves. We conclude that the boundary effect on inertial oscillations play a dominant role, and the effect of near-inertial wave is secondary. This result is quite different from previous research.

2. Clarify the difference between inertial oscillations and near-inertial waves. When I was still a PhD student several years ago, I was confused with these two stuff. Many research just mentioned them as near-inertial motions. Through comparison of these simple simulations with and without vertical stratification, I am right now very clear with their differences. In shelf seas, currents associated with inertial oscillations have opposite phases between the upper and lower layers. This property is very similar to the mode-1 structure of internal wave. Thus many research mistakenly related it to near-inertial internal waves. In our simulations, we can see this vertical structure is mainly induced by inertial oscillations (the case without stratification). The inclusion of near-inertial internal waves only produces a slight tilting of thermocline. We are sure such a comparison is valuable and easy to understand, especially for new researchers who are interested in the near-inertial motion, though this comparison is very simple.

3. Explore more detail about the two-layer structure of inertial oscillations. Yes, this feature is due to the condition of zero crossing flow toward the land boundary. Some study briefly mentioned the role of barotropic waves. But it is not clear how the barotropic
wave evolves to induce this structure. We checked out many references, but none has clarified this issue in detail. For such a fundamental property of inertial oscillation in shelf seas, it deserve a further study and a detailed interpretation. In our study, we give details on this process, many of which have not be shown in previous study. A new point is about the importance of the feedback between the barotropic wave and inertial currents.

4. Give a more physical interpretation of missing of even modes of seiches. About this missing of even modes, we do not find any other references besides Csanady's papers. Csanady got this result by using the Theorem of Residue during solving the equation with Laplace transform. It is a perfect result which is consistent with the simulation. However, it is purely mathematic. We still do not understand physically why the missing occurs. Thus, we give such an interpretation, which is more physical and easy to bear in mind. Actually, this property of seiches was not expected by us. We aimed to investigate the near-inertial motions, but just found it out and tried to understand it. As the seiches process is in a system with near-inertial motions, they are put together.

More response to specific comments :

1. 2m vertical seems quite coarse for the study. RE: We will run a simulation with 1 m in vertical to see the difference.

2. How long are the simulations? RE: 200 hours. We will add this to the manuscript.

3. Why the chosen magnitude of the wind? RE: The speed of 20 m/s is a normal value during passage of storm or clone, and is able to generate significant inertial oscillations and near-inertial internal waves.

4. It must also be noted that the barotropic Rossby radius is almost 500 km for the current set-up, so there will be interactions within the basin which must be quantified. RE: We will investigate this process and see how it works.

5. L223-224: show the computation and use SI units. RE: We will provide the computation and use SI units.

6. Appendix: this is textbook stuff and can be omitted. Instead a reference can be added. RE: Although this is quite textbook like, we think it will make the manuscript more readable if the reader does not need to look up the text book.

―――――――――――――――――――――

---

## Editor Comment (EC1) · N. C. Wells (Editor) · 2 Aug 2017

I agree with the referee that the appendix is unnecessary, and should be replaced by an appropriate reference to a textbook.

———————————————

---

## Editor Decision (ED1)

We are very appreciated for comments from two referees, which help us to improve this manuscript much. Our responses are italic. And the manuscript with tracked changes is attached behind.

**Anonymous Referee #1**

Aim of study is to investigate the absence of hypothesised even numbered modes of osculation within standing wave systems - called Seiches. Using MITGCM for an idealized basin 600km long (60m water depth) this is studied. The motivation, hypothesis and conclusion appears to be unclear, and needs to be stated; Further, the writing style also needs to be improved and much more detail and confidence in reported results are needed in my opinion. For example, a conclusion is needed in which the authors should state why study this phenomena (Why is this important?)? What was found? What this means for the scientific community (and stake holders)?

RE: This study aims to explore properties of near-inertial motions. We included the absence of even mode seiches because the seiches process occurs together with near-inertial motions. Actually they are very different topics. After a careful consideration, we decide to remove this section about the seiches and focus on discussing the near-inertial motion. Instead we add a new section (Section 5) to study the dependence of near-inertial motions on the water depth. The abstract, introduction and conclusion have been modified to make things much clearer. Please see the mark-up version of the manuscript.

Below are some further suggested changes to the text, but this is not a full list. Suggested improvements to text: Abstract - add more detail to explain why this study is about, why important, what was one and what this means. RE: Accepted and revised.

L25 - what kind of basin? an idealised rectangular? cer-tainly appears large enough for Coriolis (inertial oscillations) How was it studied? (i.e. a model). RE: *Added*.

L26 - what is "vertical stratification" I think you mean "stratified" RE: *Revised*.

L27 what are "even modes" explain further please to aid reader L53 (and throughout document)- please add space between references - for example here, "2016;Webster" should be "2016; Webster" L55 "motions(D'.." please add space and check throughout document

RE: These have been revised all through the manuscript.

L90 - 91. Nice clear aim, but why is this important? What is your hypothesis? Please be clear. RE: *Added*.

L100 - So you use the MITGCM model. Please add much more detail about this model both here and in the introduction. Why use this model? How do you know the model is correct? What are the discretised equations? How is wind stress parameterized into the model? Are the boundary conditions non-slip? What density of water is assumed? Furthermore, how can you be confident of the model results? I am sure this is a classical problem and could be compared to other models and theory for example (i.e. I could be mean here and ask if you used a different model would you get the same result?). Lastly, what stratification is used in the second case? what temperature, at what depth is thermocline?

RE: All these details have been added in the Section 2.

**Anonymous Referee #2**

The authors set out to describe inertial oscillations in a lake (or a cross section of an embayment - I'm not quite sure which). We are never told why they do this, or what the overall aim/hypothesis is, or what new knowledge we are going to get. The writing is sketchy and needs to be improved. However, at the end of the day, I don't think there is much new in this study, and I will outline why below.

RE: This study aims to explore properties of near-inertial motions. We included the absence of even mode seiches because the seiches process occurs together with near-inertial motions. Actually they are very different topics. After a careful consideration, we decide to remove this section about the seiches and focus on discussing the near-inertial motion. Instead we add a new section (Section 5) to study the dependence of near-inertial motions on the water depth. For detail, the purpose of this study is:

1. Discuss the cross-shelf distribution of near-inertial energy. As mentioned in the Introduction (Lines 66-76), observations show the near-inertial energy is maximum near the shelf break, from which it declines onshore and offshore. There has been a couple of research on discussing it, but no common interpretation is reached. Nicholls et al. (2012) simulated the near-inertial motion in the Caspian Sea, and found that the decrease of near-inertial energy depends on the distance from the coast, rather than correlated with the water depth. By solving the analytical model with a constant water depth, Pettigrew (1981) argued the

near-inertial wave is responsible for the decline of near-inertial energy near the coast. Therefore, we set up a simulation with a constant water depth to explore their ideas. The water depth is shallow (60 m) as in the shelf seas. The basin is chosen to be very wide (60 km) to guarantee that the near-inertial waves generated at one end do not which the other end during simulated duration. In our simulation, a gradual offshore increase of near-inertial energy is present in the case with only inertial oscillations. A small peak of near-inertial energy can be seen at ~50 km offshore in the case including the near-inertial waves. We conclude that the boundary effect on inertial oscillations play a dominant role, and the effect of near-inertial wave is secondary. This result is quite different from previous research.

- 2. Clarify the difference between inertial oscillations and near-inertial waves. When I was still a PhD student several years ago, I was confused with these two stuff. Many research just mentioned them as near-inertial motions. Through comparison of these simple simulations with and without vertical stratification, I am right now very clear with their differences. In shelf seas, currents associated with inertial oscillations have opposite phases between the upper and lower layers. This property is very similar to the mode-1 structure of internal wave. Thus many research mistakenly related it to near-inertial internal waves. In our simulations, we can see this vertical structure is mainly induced by inertial oscillations (the case without stratification). The inclusion of near-inertial internal waves only produces a slight tilting of thermocline. We are sure such a comparison is valuable and easy to understand, especially for new researchers who are interested in the near-inertial motion, though this comparison is very simple.
- 3. Explore more detail about the two-layer structure of inertial oscillations. Yes, this feature is due to the condition of zero crossing flow toward the land boundary. Some study briefly mentioned the role of barotropic waves. But it is not clear how the barotropic wave evolves to induce this structure. We checked out many references, but none has clarified this issue in detail. For such a fundamental property of inertial oscillation in shelf seas, it deserve a further study and a detailed interpretation. In our study, we give details on this process, many of which have not be shown in previous study. A new point is about the importance of the feedback between the barotropic wave and inertial currents.

More response to specific comments :

2m vertical seems quite coarse for the study.
 RE: *Simulations are reset and run with 1 m in vertical.*

2. How long are the simulations? RE: 200 hours. Added.

3. Why the chosen magnitude of the wind?RE: *The speed of 20 m/s is a normal value during passage of storm or clone, and is*

able to generate significant inertial oscillations and near-inertial internal waves.

- 4. It must also be noted that the barotropic Rossby radius is almost 500 km for the current set-up, so there will be interactions within the basin which must be quantified.
- RE: The width of the basin has been modified to be 300 km.
- 5. L223-224: show the computation and use SI units.

RE: The computation is just a quite simple estimation method for internal waves for a two-layer stratification. *Km/h* is more useful in our description.

6. Appendix: this is textbook stuff and can be omitted. Instead a reference can be added.

RE: Deleted.

[revised manuscript text omitted]

| 117 | $\frac{30 \text{ grids in the vertical } (z; \Delta z = 2 \text{ m})}{2 \text{ m}}$ . The water depth is uniform $\frac{(60 \text{ m})}{2}$ , with east and west sides   |
| 118 | boundaries being elosed (land)land boundaries. The vertical and horizontal eddy viscosities are assumed                                                                  |
| 119 | constants as $5 \times 10^{-4}$ m 2 /s and 10 m 2 /s, respectively. The Coriolis parameter is $5 \times 10^{-5}$ s -1 (at a latitude of |
| 120 | N). The bottom boundary is no-slip. The model is forced by a spatially uniform wind which is                                                                             |
| 121 | kept westward and increases from 0 to $0.73 \text{ N/m}^2$ (corresponding to a wind speed of 20 m/s) for the first                                                       |
| 122 | three hours and then suddenly stops. The model runs for 200 hours in total, with a time step of 4 seconds.                                                               |
| 123 | The first case has no vertical stratification is homogeneous, while the second one has a stratification of                                                               |
| 124 | two-layer structure initially. For the stratified case, #the temperature is 20°C in the upper layer (-30                                                          |
| 125 | m <z<0), (-60="" 15°c="" and="" constant,="" density="" in="" is="" is<="" layer="" lower="" m).="" m<z<-30="" salinity="" so="" th="" the=""></z<0),>                   |
| 126 | determined by the temperature.                                                                                                                                           |
| 127 | Except stratification all settings of these two cases are the same,                                                                                                      |
| 128 | 3. Inertial oscillation Results without vertical stratification                                                                                                          |
| 129 | 3.— The first case is without the presence of vertical stratification. Thus near-inertial internal wave is                                                        |

5

130 absent, and the near-inertial motion is pure inertial oscillations

**带格式的:** 字体: (默认) Times New Roman, 10 磅, 加 粗 131 For the first case without vertical stratification, seiches and inertial oscillations are two dominant 132 processes. 133 3.1 Seiches 134 Due to the westward flow driven by the wind, the water level goes up at the west coast and down at the 135 east coast initially (Fig 1). A wave front propagates from each end at the speed of the barotropic wave 136  $(\sqrt{gH} = 24 \text{ m/s or } 87 \text{ km/h})$ . As the wind stress and the water depth are uniform across the basin, the 137 elevation at the west is antisymmetric to that at the east (i.e. with the same amplitude but opposite phase). 138 The spectra of elevations is shown in Fig 2. At the inertial frequency, the elevation energy is slightly 139 increased. The most energetic peak is at the first mode of seiches, which is slightly biased by the earth 140 rotation effect. With the rotation, the wave frequency for each mode of seiche is given by (see the 141 Appendix):

142
$$-\frac{\omega_n^2 - f^2 + \frac{n^2 \pi^2 g H}{L^2}}{-(2)}$$
 (2)

where *f* is the inertial frequency, *n* the mode number, *g* the gravity acceleration, *H* the water depth, and *L* the basin width. As in most cases the horizontal scale of a closed basin is relatively small (<200 km),</li>
the second term on the right hand side of Eq. (2) is much greater than the inertial frequency term, thus
the rotation effect is usually negligible. Here due to a large basin width (600 km), the rotation effect is
obvious.-

148 The energy of the first mode is minimal at the middle of the basin (i.e., x=300 km) and maximal at two 149 boundaries. The second mode energy is almost negligible. The third mode, which has three nodes, is 150 much more energetic than the second mode. The fourth mode vanishes, while the fifth mode can be seen 151 with five nodes although it has relative low energy. In a word, the even modes are absent. In the real 152 world, due to irregular topography, there is uncertainty in computing the exact period of each mode, and 153 the research on higher modes is limited. Csanady (1973, using the Laplace transform) found the even 154 modes of seiches absent. Here we propose an alternative way combining physics and mathematics to interpret this phenomenon. 155

6

156 As derived from the appendix, the elevation of a seiche in a closed basin can be given by

---

## Author Response (AR2)

We are appreciated for comments from two referees and the editor, which help us to improve this manuscript much. All the comments from referees and the editor and our responses (italic) are listed one by one below. And the manuscript with tracked changes is attached afterwards.

**Anonymous Referee #1**

Aim of study is to investigate the absence of hypothesised even numbered modes of osculation within standing wave systems - called Seiches. Using MITGCM for an idealized basin 600km long (60m water depth) this is studied. The motivation, hypothesis and conclusion appears to be unclear, and needs to be stated; Further, the writing style also needs to be improved and much more detail and confidence in reported results are needed in my opinion. For example, a conclusion is needed in which the authors should state why study this phenomena (Why is this important?)? What was found? What this means for the scientific community (and stake holders)?

RE: *This study aims to explore properties of near-inertial motions. We included the absence of even mode seiches because the seiches process occurs together with near-inertial motions. Actually they are very different topics. After a careful consideration, we decide to remove this section about the seiches and focus on discussing the near-inertial motion. Instead we add a new section (Section 5) to study the dependence of near-inertial motions on the water depth. The abstract, introduction and conclusion have been modified to make things much clearer. Please see the mark-up version of the manuscript.*

Below are some further suggested changes to the text, but this is not a full list. Suggested improvements to text: Abstract - add more detail to explain why this study is about, why important, what was one and what this means.
RE: *Accepted and revised.*

L25 - what kind of basin? an idealised rectangular? cer-tainly appears large enough for Coriolis (inertial oscillations) How was it studied? (i.e. a model).
RE: *Added.*

L26 - what is "vertical stratification" I think you mean "stratified"
RE: *Revised.*

L27 what are "even modes" explain further please to aid reader L53 (and throughout document)- please add space between references - for example here, "2016;Webster" should be "2016; Webster" L55 "motions(D'.." please add space and check throughout document
RE: *These have been revised all through the manuscript.*

L90 - 91. Nice clear aim, but why is this important? What is your hypothesis? Please be clear.
RE: *Added.*

L100 - So you use the MITGCM model. Please add much more detail about this model both here and in the introduction. Why use this model? How do you know the model is correct? What are the discretised equations? How is wind stress parameterized into the model? Are the boundary conditions non-slip? What density of water is assumed? Furthermore, how can you be confident of the model results? I am sure this is a classical problem and could be compared to other models and theory for example (i.e. I could be mean here and ask if you used a different model would you get the same result?). Lastly, what stratification is used in the second case? what temperature, at what depth is thermocline?
RE: *All these details have been added in the Section 2.*

**Anonymous Referee #2**

The authors set out to describe inertial oscillations in a lake (or a cross section of an embayment – I'm not quite sure which). We are never told why they do this, or what the overall aim/hypothesis is, or what new knowledge we are going to get. The writing is sketchy and needs to be improved. However, at the end of the day, I don't think there is much new in this study, and I will outline why below.

RE: *This study aims to explore properties of near-inertial motions. We included the absence of even mode seiches because the seiches process occurs together with near-inertial motions. Actually they are very different topics. After a careful consideration, we decide to remove this section about the seiches and focus on discussing the near-inertial motion. Instead we add a new section (Section 5) to study the dependence of near-inertial motions on the water depth. For detail, the purpose of this study is:*

1. *Discuss the cross-shelf distribution of near-inertial energy. As mentioned in the Introduction (Lines 66-76), observations show the near-inertial energy is maximum near the shelf break, from which it declines onshore and offshore. There has been a couple of research on discussing it, but no common interpretation is reached. Nicholls et al. (2012) simulated the near-inertial motion in the Caspian Sea, and found that the decrease of near-inertial energy depends on the distance from the coast, rather than correlated with the water depth. By solving the*

*analytical model with a constant water depth, Pettigrew (1981) argued the near-inertial wave is responsible for the decline of near-inertial energy near the coast. Therefore, we set up a simulation with a constant water depth to explore their ideas. The water depth is shallow (60 m) as in the shelf seas. The basin is chosen to be very wide (300 km) to guarantee that the near-inertial waves generated at one end do not reach the other end during simulated duration. In our simulation, a gradual offshore increase of near-inertial energy is present in the case with only inertial oscillations. A small peak of near-inertial energy can be seen at ~50 km offshore in the case including the near-inertial waves. We conclude that the boundary effect on inertial oscillations play a dominant role, and the effect of near-inertial wave is secondary. This result is quite different from previous research.*

2. *Clarify the difference between inertial oscillations and near-inertial waves. When I was still a PhD student several years ago, I was confused with these two stuff. Many research just mentioned them as near-inertial motions. Through comparison of these simple simulations with and without vertical stratification, I am right now very clear with their differences. In shelf seas, currents associated with inertial oscillations have opposite phases between the upper and lower layers. This property is very similar to the mode-1 structure of internal wave. Thus many research mistakenly related it to near-inertial internal waves. In our simulations, we can see this vertical structure is mainly induced by inertial oscillations (the case without stratification). The inclusion of near-inertial internal waves only produces a slight tilting of thermocline. We are sure such a comparison is valuable and easy to understand, especially for new researchers who are interested in the near-inertial motion, though this comparison is very simple.*

3. *Explore more detail about the two-layer structure of inertial oscillations. Yes, this feature is due to the condition of zero crossing flow toward the land boundary. Some study briefly mentioned the role of barotropic waves. But it is not clear how the barotropic wave evolves to induce this structure. We checked out many references, but none has clarified this issue in detail. For such a fundamental property of inertial oscillation in shelf seas, it deserve a further study and a detailed interpretation. In our study, we give details on this process, many of which have not be shown in previous study. A new point is about the importance of the feedback between the barotropic wave and inertial currents.*

More response to specific comments :

1. 2m vertical seems quite coarse for the study.
RE: *Simulations are reset and run with 1 m in vertical.*

2. How long are the simulations?
RE: *200 hours. Added.*

3. Why the chosen magnitude of the wind?

RE: *The speed of 20 m/s is a normal value during passage of storm or clone, and is able to generate significant inertial oscillations and near-inertial internal waves.*

4. It must also be noted that the barotropic Rossby radius is almost 500 km for the current set-up, so there will be interactions within the basin which must be quantified.
RE: *The width of the basin has been modified to be 300 km.*

5. L223-224: show the computation and use SI units.
RE: *The computation is just a quite simple estimation method for internal waves for a two-layer stratification. Km/h is more useful in our description.*

6. Appendix: this is textbook stuff and can be omitted. Instead a reference can be added.
RE: *Deleted.*

**From the topic editor**

The paper is interesting comparison between theoretical work and a non-linear numerical model ( MIT model) on inertial currents and waves. The revised version is clear presentation of the model results and gives some insight into to the differences between the model and published theory.

However, the reviewers have commented on the fact that the model is only 2 dimensional with lateral boundaries. It is therefore only applicable to a lake or an embayment on the shelf. It would be interesting to look at an extension to where one boundary was open to replicate the response to a transverse continental shelf.

The revised version can be published with minor revisions. I have attached a version which has corrections to the English.

RE: *We are very glad that you think it is an interesting comparison between theory and simulation, which is our motivation. It is definitely a good direction to run some new cases with open boundaries.*

1. Line 33, 'increasing' replaced by 'higher'.
RE: *Accepted and revised.*

2. Line 42, delete 'formed'.
RE: *Accepted and revised.*

3.  Line 80, offshore is not the coast. I do not understand this highlighted phrase.
RE: *It means the inertial energy is leaked downward to the lower layer and offshore, thus the inertial energy near the coast is reduced. This sentence has been slightly changed to make it clearer.*

4.  Line 81, 'seems complicated' replaced by 'is not clear'.
RE: *Accepted and revised.*

5.  Line 85, 'of' replaced by 'between'.
RE: *Accepted and revised.*

6.  Line 126, you should either mention the corresponding density difference between the top and bottom layer, or give the thermal expansion coefficient,   because this controls the long baroclinic wave speed.
RE: *We clarify this by adding a sentence: The salinity is constant, and the density is linearly determined by the temperature, with an expansion coefficient of $2\times10^{-4}\ ^{\circ}C^{-1}$.*

7.  Line 127, this seems contradict the previous sentence.
RE: *Deleted as suggested.*

8.  Line 179, period of the first mode seiche?
RE: *It is 6.9 hours which is added.*

9.  Line 204, 'great' replaced by 'large'.
RE: *Accepted and revised.*

10. Line 219, 'correlated' replaced by 'related'.
RE: *Accepted and revised.*

11. Line 254, use hours as well to be consistent with line 248.
RE: *As circle per day is a more common and intuitive unit, we think it better to keep using cpd.*

12. Line 264, It would be worth giving the values for the barotropic and baroclinic Rossby radii in the method section.
RE: *Yes, we give these values in the method section (end of the Section 2).*

13. Line 295, 'overpasses' replaced by 'exceeded', and 'sum' replaced by 'total'.
RE: *Accepted and revised.*

14. Line 302, 'suppressing' replaced by 'suppression'.
RE: *Accepted and revised.*

15. Line 335, 'low' replaced by 'lower'.
RE: *Accepted and revised.*

16. Line 341, this is a comparison of theory with a numerical model, and therefore you need explain why the features are more realistic. Where are the observations to test the theory or model against? The model is more complex than the theory. For example the effect of the non-linear terms and this is perhaps.
RE: *Yes, it is mainly due to nonlinearity of the model which brings the difference. We add a sentence to clarify this. Actually, I do not find any references which use observation to test this theoretical feature about the varying frequencies of the Rossby adjustment. It is a good idea to try it.*

17. Line 364, 'makes' replaced by 'has'.
RE: *Accepted and revised.*

There are many places missing 'a' or 'the' as the editor pointed out. We have accepted and revised them all. Thanks a lot for such a careful review.

[revised manuscript text omitted]